# Strengthening individual and family resilience against leprosy-related discrimination: A pilot intervention study

**Anna T. van't Noordende**[1,2]*, **Zoica Bakirtzief da Silva Pereira**[3], **Pritha Biswas**[4], **Mohammed Ilyas**[5], **Vijay Krishnan**[6], **Jayaram Parasa**[4], **Pim Kuipers**[7]

**1** NLR, Amsterdam, The Netherlands, **2** Erasmus MC, University Medical Center Rotterdam, Rotterdam, The Netherlands, **3** Federal University of Santa Maria, Santa Maria, Brazil, **4** Lepra Society, Hyderabad, India, **5** Hyderabad Leprosy Control & Health Society, Hyderabad, India, **6** Fontilles, Hyderabad, India, **7** The Hopkins Centre, Griffith University, Queensland, Australia

* a.vt.noordende@nlrinternational.org

**Data Availability Statement:** All data underlying the findings are available from the Infolep website:

## Abstract

### Background

Leprosy and leprosy-related stigma can have a major impact on psychosocial wellbeing of persons affected and their family members. Resilience is a process that incorporates many of the core skills and abilities which may enable people to address stigma and discrimination. The current study aimed to develop and pilot an intervention to strengthen individual and family resilience against leprosy-related discrimination.

### Methodology

We used a quasi-experimental, before-after study design with a mixed methods approach. The 10-week family-based intervention was designed to strengthen the resilience of individuals and families by enhancing their protective abilities and capacity to overcome adversity. The study was conducted in two sites, urban areas in Telangana state, and in rural areas in Odisha state, India. Persons affected and their family members were included using purposive sampling. Two questionnaires were used pre-and post-intervention: the Connor-Davidson Resilience Scale (CD-RISC, maximum score 100, with high scores reflecting greater resilience) and the WHOQOL-BREF (maximum score of 130, with higher scores reflecting higher quality of life). In addition, semi-structured interviews were conducted post-intervention. Data were collected at baseline, a few weeks after completion of the intervention, and in the Odisha cohort again at six months after completion. Paired t-tests measured differences pre- and post- intervention. Qualitative data were thematically analysed.

### Findings

Eighty participants across 20 families were included in the study (23 persons affected and 57 family members). We found a significant increase in CD-RISC scores for persons affected and family members from Odisha state (baseline 46.5, first follow-up 77.0, second follow-up 70.0), this improvement was maintained at six-month follow-up. There was no

https://www.infontd.org/sites/default/files/2021-02/Resilience study database.txt.

**Funding:** This study is funded by the Leprosy Research Initiative Foundation (LRI, leprosyresearch.org) under project number 706.18.46. The funders had no role in study design, data collection and analysis, decision to publish, or preparation of the manuscript.

**Competing interests:** The authors have declared that no competing interests exist.

increase in CD-RISC scores post-intervention among participants from Telangana state. WHOQOL-BREF scores were significantly higher at follow-up for persons affected in both states, and for family members in Odisha state. No families dropped out of the study. In the qualitative feedback, all participants described drawing benefit from the programme. Participants especially appreciated the social dimensions of the intervention.

## Conclusion

This pilot study showed that the 10-week family-based intervention to strengthen resilience among persons affected by leprosy and their family members was feasible, and has the potential to improve resilience and quality of life. A large-scale efficacy trial is necessary to determine the effectiveness and long-term sustainability of the intervention.

### Author summary

The current study aimed to develop and pilot an intervention to strengthen individual and family resilience against leprosy-related discrimination. The study was conducted in two sites, urban areas in Telangana state, and in rural areas in Odisha state, India. We assessed resilience and quality of life with two questionnaires (the CD-RISC and WHO-QOL-BREF) pre-and post-intervention. In addition, semi-structured interviews were conducted post-intervention. Data were collected at baseline, a few weeks after completion of the intervention, and in Odisha state again at six months after completion. Eighty participants across 20 families were included in the study (23 persons affected and 57 family members). We found a significant increase in resilience scores for persons affected and family members from Odisha state, this improvement was maintained at six-month follow-up. There was no increase in resilience scores post-intervention among participants from Telangana state. Quality of life scores were significantly higher at follow-up for persons affected in both states, and for family members in Odisha state. Participants especially appreciated the social dimensions of the intervention. This pilot study showed that the 10-week family-based intervention to strengthen resilience among persons affected by leprosy and their family members was feasible, and has the potential to improve resilience and quality of life.

## Introduction

Leprosy is a chronic infectious disease caused by *Mycobacterium leprae* [1]. Although curable by a combination of drugs, the disease may damage the peripheral nerves and skin, which, especially if diagnosed late, can result in impairment [1]. Such impairments and other factors (including social myths, attitudes cultural and religious beliefs and fear), may lead to persons affected by leprosy being stigmatised [2,3]. Social stigma may impact the emotions, thoughts, behaviour and relationships of persons who are stigmatised [4]. Not surprisingly, leprosy has been associated with anxiety disorders, depression, suicide (attempts), mental distress and emotions such as fear and shame, low self-esteem and reduced quality of life [5]. Close contacts of persons affected, such as family members and friends, may also be negatively impacted by such social and psychological consequences of the disease [5].

Though leprosy can have a major impact on psychosocial wellbeing [5], there is also evidence that persons affected by leprosy can overcome experiences of discrimination and exclusion [6]. Studies indicate that people who have faced severe stigmatisation and have gone on to overcome this adversity demonstrate numerous dimensions of resilience [7,8]. They also tend to show normal or high levels of mental wellbeing despite exposure to psychological or physical adversity [9]. It would appear that resilience is a process that incorporates many of the core skills and abilities which may be required to address stigma and discrimination.

Resilience can be understood as a dynamic and complex process (as well as outcome) of successfully adapting to difficult or challenging life experiences, especially through mental, emotional and behavioral flexibility and adjustment to external and internal demands [10,11]. Many factors can contribute to how well or poorly people adapt to adversity, including how they perceive and engage with the environment, their social resources and their use of coping strategies [12–15]. Many of these "protective" or adaptive factors can be learned and strengthened, such as emotional regulation, self-efficacy, support seeking behaviour, communication skills, problem solving abilities and engaging in a supportive environment [12–15]. In the face of stressors, these protective factors are said to modify the individual's or groups' response to adversity, thereby reducing the likelihood of negative outcomes [15].

Evidence-based approaches to building resilience within families have been described [12], including where a family member has health and disability problems [16], or is stigmatised and in resource-poor settings [17]. Versions of these approaches are now also conducted as brief interventions [18], addressing family stress, conflict, cohesion, adaptation, working through adversity, beliefs and spirituality, and broader communication beyond the family. Unfortunately, most of them are highly resource intensive, with even the brief interventions requiring multiple counselling sessions with each family over many months. In addition, most are not designed for neglected tropical disease (NTD) services such as leprosy programmes. The key challenge is how to make these highly strategic interventions more suited to the realities of treatment, rehabilitation and other services, and thereby more accessible to people in leprosy-endemic countries.

The current study aimed to develop and pilot an intervention to strengthen individual and family resilience against leprosy-related discrimination. It is based on a scoping review [19] to determine principles of evidence-based interventions to strengthen personal and family resilience, as well as a qualitative exploration on sources of strength and resilience [20].

## Methods

### Ethics statement

Ethical approval was obtained from the Technical Advisory Group and the Institutional Ethics Committee of the Lepra-Blue Peter Public Health and Research Centre (BPHRC). Written informed consent was obtained from all participants prior to data collection.

### Study design and study area

We used a quasi-experimental, before-after study design using mixed methods data collection. The study was conducted in urban areas in Hyderabad district in Telangana state and in rural areas in Subarnnapur, Nabarangpur and Koraput districts in Odisha state, India. In 2019, the prevalence of leprosy was 1.45 per 10,000 population in Odisha state and 0.62 per 10,000 population in Telangana state (State NLEP report, 2019–2020).

## Description of the intervention

The intervention was designed to strengthen the resilience of families struggling with leprosy-related discrimination, by strengthening their protective abilities and capacity to overcome adversity. The contents of the intervention were based on a recent scoping review [19], a qualitative exploration on sources of strength and resilience [20], determinants of individual, family and community resilience among families in low- and middle-income contexts [21] and the family resilience framework [12]. An overview of the content of each session and corresponding resilience factors can be found as supporting information file (S1 Text).

Our scoping review identified a number of key process-related factors: the intervention should have more than one session, it should be spread over several weeks, it should ideally involve persons affected themselves in planning or executing the intervention, it should be targeted to fit the audience and it should include multiple intervention components [19]. The intervention to strengthen resilience was therefore designed to have ten weekly sessions in a family-based format. Each session adopted an action learning and problem-solving approach to the activities design. In addition, a separate session about rights was added, since our scoping review identified a human rights foundation as an important component [19]. In addition, our qualitative exploration on sources of strength and resilience identified the importance of supporting family and social relationships, providing accurate information about leprosy, and acknowledging spiritual beliefs [20]. We therefore added a separate session about knowledge about leprosy, and also integrated these other concerns. The scoping review endorsed the importance of providing correct knowledge about the stigmatized condition to empower the individual or group to challenge misconceptions about their condition.

While it was not logistically possible to include persons affected in executing the intervention for this pilot, the study team included a person affected (ZBSP), and the intervention was developed with reference to the ILEP Advisory Panel, comprising women and men affected by leprosy. In addition, a three-day interactive training workshop was conducted in Hyderabad by three authors with the staff members who had been selected to deliver the intervention. In this workshop the assessments were presented and discussed, and the intervention content and resource materials were discussed and refined.

In order to make the concept of resilience more tangible to the participants (largely poor Indian families), bamboo was chosen as emblematic of the intervention pictorially and as a catch-phrase: 'being strong and bouncing back like the bamboo in a storm'. We identified four main themes: strong roots of the bamboo plant (knowledge, sessions 1–2), strong trunk (thoughts and behaviour, sessions 3–5), strong branches and leaves (rights and spirituality, sessions 6–7), and strong soil (relationships and social support, sessions 8–10).

## Participants and sampling procedure

Persons who had been treated for leprosy ('persons affected') and their family members were included in the study. Participants lived in urban slum areas of Hyderabad, and in rural tribal areas of Subarnapur, Nabarangpur and Koraput districts of Odisha. Family members included in the study were all people in the same household as the identified person affected by leprosy, they were related by blood or by marriage. Persons below the age of 18 and those unwilling or unable to give informed consent were excluded. Those living in single person households were also excluded.

Participants were purposively sampled; contacted through the networks of Lepra Society (Odisha cohort) and Hyderabad Leprosy Control & Health Society (HLCHS) (Telangana cohort). Identified families were visited in their home to invite them to participate and to

check whether they met the inclusion criteria. Sampling also sought a mix of women and men affected by leprosy, of different ages.

According to Hertzog [22], if the aim of the pilot study is to demonstrate intervention efficacy in a single group, a sample size of 20–25 is adequate. A sample size of 10–20 participants per group is adequate when trying to determine group differences [22]. Since this was a proof of concept (pilot) study in which we wanted to explore outcomes of a trial intervention, but also wanted to determine if there are any differences between rural and urban areas, we aimed to include families from rural as well as urban areas.

### Data collection

**Data collection tools.**    Demographic information gathered included sex, age, education, occupation, religion, role in the family, and for persons affected, also included disability grade and year of diagnosis. We used two validated questionnaires: the Connor-Davidson Resilience Scale (CD-RISC) and WHOQOL-BREF. In addition, semi-structured interviews to assess participant satisfaction were conducted with each family post-intervention, with facilitators taking notes of the responses of participants to each question.

The CD-RISC was used to assess (protective factors related to) resilience. The scale comprises of 25 items, each item can be rated from 0 ('not true at all') to 4 ('true nearly all the time'). A total score of 100 can be obtained, with high scores reflecting greater resilience [23]. The CD-RISC has been validated in Urdu [24] and Hindi [25]. The CD-RISC has not been validated in Odia language (spoken in Odisha), so we used an external language expert to translate it (using both English and Hindi versions as reference). Local bilingual supervisors checked and corrected this version, which was subsequently cross checked by bilingual implementing staff. Supervisors and implementing staff then developed local protocols for administering the scale in the local context.

The WHOQOL-BREF was used to assess quality of life, a concept that is closely linked to well-being and broadly encompasses someone's perception of how 'good' several aspects of their life are [26]. The WHOQOL-BREF comprises 26 items, each rated from 1 (negative response) to 5 (positive response). A total score can be obtained for each of the four domains: physical health (raw total score 35), psychological (30), social relationships (15) and environment (40). Question 1 (overall perception of quality of life) and question 2 (overall perception of health) are not included in the domain scores [27]. Even though there is no official total score for the WHOQOL-BREF [27], we calculated a sum score of 130 by adding the scores obtained on each of the 26 questions to be able to compare the total sum scores pre- and post-intervention. This was done after transforming the scores on negatively framed questions (question 3, 4 and 26). Higher scores on the WHOQOL-BREF denote higher quality of life. The WHOQOL-BREF has been validated in Urdu [28], Hindi [29] and Odia [30].

In addition, staff records were maintained documenting session dates, number of participants and staff comments. Both the intervention and the interviews were conducted by trained facilitators working in leprosy for several years, in participants' homes.

**Phases of follow-up.**    Three rounds of data collection were completed in Odisha state: the baseline assessment was conducted in October 2019, the first follow-up in February 2020 (one week after the intervention had been completed) and the second follow-up in August 2020 (six months after the intervention had been completed). Two rounds of questionnaires were conducted in Telangana state: the baseline assessment was conducted in November 2019 and the follow-up in April 2020 (three and a half weeks after the intervention had been completed). A subsequent follow-up interview was conducted by telephone in Telangana state in October

2020 (six months after the intervention had been completed). Telephone administration was used due to the high number of people affected by COVID-19 in the area.

## Data analysis

Quantitative data were collected on paper forms and entered in a database created using Epi Info. The data were analysed using IBM SPSS Statistics 24. Differences between participants from Odisha and Telangana state were evaluated using the Mann-Whitney U test for continuous variables, X2 statistics for categorical variables and Fisher's Exact test for categorical variables for which the expected values in one of the cells of the contingency table was less than five. Frequencies and percentages were calculated to depict demographic information of the participants. Median total scores and the interquartile range (IQR) of the CD-RISC and WHOQOL-BREF were calculated for the different participant groups to summarize resilience and quality of life scores pre- and post-intervention. The Wilcoxon signed-rank test was performed to check whether the scores were significantly different pre- and post- intervention. In addition, we created a new variable that contained the absolute difference between the baseline and follow-up scores of the CD-RISC and WHOQOL-BREF and used bootstrapped stepwise multivariate linear regression with backward elimination to see if there were associations between these variables and the other variables in our dataset (sex, age, participant type, occupation, education, role in the family, and in Telangana state religion also). Bootstrapping was performed to correct for non-normality of the data. Variables were included in the model if they had a p-value of <0.2 in univariate analysis. We made separate models for each state. Statistical significance level was set a priori at $p < 0.05$. All data were anonymised before data analysis.

Qualitative data (detailed notes of family responses to interview questions, as well as staff records of weekly meetings) were thematically analysed, identifying key elements of the intervention described as beneficial by participants.

## Results

### Demographic information

A total of 80 participants were included in the study. A little over half of the participants (n = 41, 51%) were from urban areas in Telangana state. The median age of the participants was 35 years. Half of the participants had not had any (formal) education (n = 40, 50%). About one third had paid work (n = 24, 30%), and a third were unemployed (n = 23, 29%). While all participants in Odisha state were Hindu (n = 39, 100%), the participants in Telangana state were either Muslim (n = 21, 51%) or Hindu (n = 20, 49%). An overview of all participant characteristics can be found in Table 1.

Persons affected (n = 23, 29%) and family members (n = 57, 71%) were included. Most persons affected (n = 16, 70%) had grade 2 disabilities, followed by grade 0 (n = 5, 22%) and grade 1 (n = 4, 17%). Most persons affected were diagnosed between five and ten years ago (n = 14, 61%). Only one participant (n = 1, 4%) was diagnosed over ten years ago. Family members included caregivers (n = 21, 37%), breadwinners (n = 14, 25%), or heads of the household/ decision maker (n = 23, 35%).

### Family-based sessions

In most cases, about four people participated in the intervention (range 3–6). The intervention consisted of ten weekly sessions that were held in participants' homes. On average, all family members were present for eight of the ten sessions. Each session lasted on average 63 minutes

**Table 1. Demographic information of the study population.**

| | Odisha state (n = 39) | Telangana state (n = 41) | Total (n = 80) | p-value[b] |
|---|---|---|---|---|
| Age, median (interquartile range) | 35.0 (24.0–50.0) | 36.0 (27.0–59.0) | 35.0 (25.3–55) | 0.187 |
| Sex, *n* (%) | | | | 0.642 |
| Female | 22 (56.4) | 21 (51.2) | 43 (53.8) | |
| Male | 17 (43.6) | 20 (48.8) | 37 (46.2) | |
| Living area, *n* (%) | | | | 0.000 |
| Rural | 39 (100.0) | 0 (0.0) | 39 (48.8) | |
| Urban | 0 (0.0) | 41 (100.0) | 41 (51.3) | |
| Language, *n* (%) | | | | 0.000 |
| Hindi | 1 (2.6) | 19 (46.3) | 20 (25.0) | |
| Urdu | 1 (2.6) | 20 (48.8) | 21 (26.3) | |
| Odia | 37 (94.9) | 0 (0.0) | 37 (46.3) | |
| Telegu | 0 (0.0) | 18 (43.9) | 18 (22.5) | |
| Religion, *n* (%) | | | | 0.000 |
| Hindu | 39 (100.0) | 20 (48.8) | 59 (73.8) | |
| Muslim | 0 (0.0) | 21 (51.2) | 21 (26.3) | |
| Occupation at baseline, *n* (%) | | | | 0.001 |
| Paid work | 9 (23.1) | 15 (36.6) | 24 (30.0) | |
| Unemployed | 6 (15.4) | 17 (41.5) | 23 (28.8) | |
| Other[a] | 24 (61.5) | 9 (22.0) | 33 (41.3) | |
| Education, *n* (%) | | | | 0.733 |
| No or no formal education | 21 (53.8) | 19 (46.3) | 40 (50.0) | |
| Primary | 12 (30.8) | 16 (39.0) | 28 (35.0) | |
| Secondary or higher | 6 (15.4) | 6 (14.6) | 12 (15.0) | |
| Participant type, *n* (%) | | | | 0.697 |
| Persons affected | 12 (30.8) | 11 (26.8) | 23 (28.8) | |
| Family member | 27 (69.2) | 30 (73.2) | 57 (71.3) | |

[a] Occupation 'other' included non-paid work, self-employed and retired.

[b] The tests used are the Mann-Whitney U test for continuous variables (age), X2 statistics for categorical variables (sex, occupation, education and participant type) and Fisher's Exact test for categorical variables for which the expected values in one of the cells of the contingency table was less than five (living area, language and religion).

(74 minutes in Odisha state and 51 minutes in Telangana state). In general, the duration of each session increased with family size.

## Short-term impact on resilience and quality of life

Table 2 shows the median difference in CD-RISC (resilience) and WHOQOL-BREF (quality of life) scores between the baseline and first follow-up assessment. Higher scores reflect greater resilience and higher quality of life. The increase in resilience scores is significant for persons affected and family members from Odisha. There is no significant improvement in CD-RISC scores among persons from Telangana state pre- and post-intervention. However, there is a significant improvement post-intervention when only looking at the Hindu participants from Telangana state (Wilcoxon signed-rank test, p = 0.048) In addition, there is a significant increase in quality of life scores for all participant groups pre- and post-intervention, except for family members from Telangana state (Table 2).

An overview of the median difference pre- and post-intervention per question of the CD-RISC and WHOQOL-BREF and of the difference in domain scores on the WHOQOL-BREF can be found as supporting information file (S2 Text).

In Odisha state, all domains of the WHOQOL-BREF (physical health, psychological, social relationships and environment) significantly improved post-intervention. In Telangana state, only the domains social relationships and environment significantly improved post-intervention (S2 Text).

**Table 2. Difference in baseline and first follow-up in resilience scores (CD-RISC) and quality of life scores (WHOLQOL-BREF).**

| | | Baseline Median (IQR) | Follow-up Median (IQR) | Difference (%) | p-value[a] |
|---|---|---|---|---|---|
| CD-RISC | Participants from Odisha (n = 38) | 46.5 (39.8–56.0) | 77.0 (68.0–86.0) | 30.5 (65.6) | 0.000 |
| | Persons affected from Odisha (n = 12) | 40.5 (32.3–57.0) | 75.0 (65.0–84.8) | 34.5 (85.2) | 0.002 |
| | Family members from Odisha (n = 26) | 47.5 (42.0–56.0) | 77.0 (68.0–87.0) | 29.5 (62.1) | 0.000 |
| | Participants from Telangana (n = 41) | 49.0 (46.5–51.5) | 50.0 (47.0–52.0) | 1.0 (2.0) | 0.471 |
| | Persons affected from Telangana (n = 11) | 49.0 (44.0–50.0) | 47.0 (43.0–50.0) | 2.0 (4.1) | 1.000 |
| | Family members from Telangana (n = 30) | 49.0 (47.0–53.3) | 50.0 (47.0–52.3) | 1.0 (2.0) | 0.362 |
| WHOQOL-BREF | Participants from Odisha (n = 39) | 75.0 (67.0–81.0) | 100.0 (94.0–105.0) | 25.0 (33.3) | 0.000 |
| | Persons affected from Odisha (n = 12) | 66.5 (62.5–77.3) | 97.0 (92.5–101.8) | 30.5 (45.9) | 0.002 |
| | Family members from Odisha (n = 27) | 77.0 (73.0–83.0) | 101.0 (94.0–107.0) | 24.0 (31.2) | 0.000 |
| | Participants from Telangana (n = 41) | 69.0 (62.0–73.5) | 76.0 (67.5–79.5) | 7.0 (10.1) | 0.004 |
| | Persons affected from Telangana (n = 11) | 65.0 (62.0–69.0) | 78.0 (71.0–81.0) | 13.0 (20.0) | 0.010 |
| | Family members from Telangana (n = 30) | 75.5 (65.0–79.0) | 71.5 (62.0–74.3) | 4.0 (5.3) | 0.108 |

[a] P-value of the baseline versus first follow-up scores, calculated using the Wilcoxon signed-rank test.

## Factors associated with short-term increase in scores

Some factors were associated with short-term increase on the CD-RISC and WHOQOL-BREF scales (please see S3 Text). We developed two models for each state, given the large differences in median increase on the two scales per state.

Multivariate analysis showed that participants from Odisha state, with occupation 'other' (e.g. day labourer) had significantly less improvement on the CD-RISC between baseline and the first follow-up assessment (Table 1). This model explained 16% of the variability of increase in resilience score in Odisha state. In addition, multivariate analysis showed that men had significantly more improvement on the WHOQOL-BREF between baseline and the first follow-up assessments (Table 2). This model explained 17% of the variability of increase in quality of life score in Odisha state.

Multivariate analysis of the data from Telangana state showed that participants who had another occupation than paid work and participants who were Hindu had significantly more improvement on the CD-RISC between baseline and the first follow-up assessments (Table 3). This model explained 24% of the variability of increase in resilience score in Telangana state. In addition, analysis showed that persons affected by leprosy and participants who were Hindu had significantly more improvement on the WHOQOL-BREF between baseline and the first follow-up, this model explained 50% of the variability of increase in quality of life score in Telangana state–religion alone explained 40% of the variability (r-squared 0.401).

## Impact on resilience and quality of life after six months

Participants from Odisha state underwent an additional follow-up assessment at six months post-intervention, which coincided with the COVID-19 lockdowns in India. Table 3 shows the median difference in CD-RISC and WHOQOL-BREF scores between baseline, first follow-up and second follow-up assessments in Odisha state. All median scores decreased between first and second follow-up. This decrease was significant for all subgroups on both scales, except for the resilience score of persons affected by leprosy in Odisha state. Even though the resilience and quality of life scores decreased between first and second follow-up, the scores for all subgroups on both scales remained significantly higher than baseline (Table 3).

**Table 3. Difference in baseline and first and second follow-up in resilience scores (CD-RISC) and quality of life scores (WHOQOL-BREF) in Odisha state.**

|  |  | Baseline Median (IQR) | First follow-up Median (IQR) | Second follow-up Median (IQR) | p-value baseline versus first follow-up | p-value baseline versus second follow-up |
|---|---|---|---|---|---|---|
| CD-RISC | All participants from Odisha (n = 38) | 46.5 (39.8–56.0) | 77.0 (68.0–86.0) | 70.0 (64.0–79.0) | 0.000 | 0.000 |
|  | Persons affected from Odisha (n = 12) | 40.5 (32.3–57.0) | 75.0 (65.0–84.8) | 73.5 (61.5–86.5) | 0.002 | 0.002 |
|  | Family members from Odisha (n = 26) | 47.5 (42.0–56.0) | 77.0 (68.0–87.0) | 69.0 (65.0–76.0) | 0.000 | 0.000 |
| WHOQOL-BREF | Participants from Odisha (n = 39) | 75.0 (67.0–81.0) | 100.0 (94.0–105.0) | 92.0 (87.0–98.0) | 0.000 | 0.000 |
|  | Persons affected from Odisha (n = 12) | 66.5 (62.5–77.3) | 97.0 (92.5–101.8) | 91.5 (87.5–95.0) | 0.002 | 0.002 |
|  | Family members from Odisha (n = 27) | 77.0 (73.0–83.0) | 101.0 (94.0–107.0) | 95.0 (86.0–100.0) | 0.000 | 0.000 |

[a] We used the Wilcoxon signed-rank test.

## Qualitative data

It was evident from the interview notes that participants greatly valued the 10-week program and enjoyed the practical, vignette- and story-based approach. Qualitative feedback indicated that participants enjoyed the social dimensions of the program. In feedback some noted that the project gave them more confidence to "develop relationship(s) with others" (AD:BS).

Across interview notes there were clear indications that the family-based approach enabled greater connection and social strengthening, promoting understanding and acceptance "We could discuss about health issue together with family members by this programme" (AJ:PN), and that it "brought the change among us" (BS:NN).

Nearly all participants mentioned that they appreciated and drew benefit from the visual image and repeated metaphor of 'being strong and bouncing back like the bamboo in a storm'. Interviews also reflected that participants understood that resilience was multifactorial, including health related, psychological, behavioural, family, social, rights and other dimensions. All interviewees expressed some degree of improvement in at least some of these aspects.

Across all participants, week 10 (social activity with peers), week 2 (knowledge about leprosy), week 6 (understanding your rights), week 8 (family relationships) and week 3 (positive thinking, understanding thoughts and emotions) were identified as the most beneficial. A number of participants indicated that they thought that week 7 (spirituality) could be improved.

Analysis of staff weekly notes about the 10-week program indicated that staff seemed to appreciate the detailed program which highlighted a number of issues for discussion and action. In many settings, even community agencies, leprosy services are quite narrow, focusing on treatment and follow-up. Notes indicated that for some staff, the project was an entry point to addressing a broader range of important issues for families. The manual appeared to give staff both the content and frameworks to promote discussion on a number of topics, and offer more psycho-socially oriented support. Staff seemed to like the program approach of working with families (as opposed to individuals) and used the program to incorporate a number of psycho-emotional, social, rights-based, and practical issues (pertaining to benefits and treatments) into their service provision.

## Discussion

The high prevalence and negative impact (including psychosocial impact) of leprosy-related stigma is well described in the literature [31–36]. However, there are few psychosocial and

resilience interventions designed to assist people in NTD and leprosy programs. Further, many of the existing resilience interventions are highly resource intensive and suited to Western contexts. While it is clear that resilience can be strengthened [12–15,19,37,38], the present pilot study indicates the potential of such actions in the context of leprosy-related discrimination. Findings from this study suggest that a 10-week family-based intervention to strengthen resilience among persons affected by leprosy and their family members is feasible, and has the potential to improve resilience and quality of life.

## Content of intervention

The intervention in the present study consisted of ten sessions that focused on four main themes: knowledge, thoughts and behaviour, rights and spirituality, and relationships and social support. These broad themes incorporated elements similar to those found in a recent review by Chmitorz and colleagues [39], who reviewed and evaluated 43 randomized controlled training programs to foster psychological resilience. They found that these programmes most often included cognitive restructuring, stress management, problem-solving and coping strategies.

While most components of the current intervention were based on resilience factors relevant to families in low- and middle-income contexts [12,21], the sessions about 'knowledge' and 'rights' were added based on our recent scoping review [19] and qualitative exploration of sources of strength and resilience [20]. Several studies have shown that accurate knowledge about leprosy is associated with reduced stigma [40–42]. Misconceptions about leprosy, often linked to fear of the disease and fear of transmission, can increase stigma [2,43–45]. We therefore considered knowledge a tool to help family members address and challenge misconceptions and reduce (community) stigma. Knowledge can also reduce internalised stigma, by helping increase someone's self-image. This was illustrated by Lusli and colleagues [46], who found that increased knowledge helped persons affected by leprosy to see themselves as cured and no longer infectious, rather than infectious and uncured. This enhanced their self-perception [46]. Human rights was identified as an important component to include in the intervention [19]. Persons affected by leprosy who are aware of their rights have been found to be more confident and less afraid to take initiatives [46].

Qualitative responses indicated that the social dimensions of the intervention were especially appreciated by the participants. Accordingly, there was a significant increase of scores in the domains "social relationships" and "environment" of the WHOQOL-BREF scale in both states post-intervention. Many studies have emphasized the importance of social support in fostering resilience [12,47–49]. Support for resilience is provided by family, friends, neighbours and mentors [47]. In children, peer acceptance and friendships have been found to act as moderators between family adversity and child adjustment to adversity [48]. Werner [47] found that self-esteem and self-efficacy are promoted through supportive relationships. Family relationships provide practical and emotional support in the context of discrimination [10,12]. In addition, people feel less stressed when they have good family and social support [10]. Good social relationships are vital for resilience [12] and we consider it a key component of the current intervention.

## Family-based format

The present study used a family-based format. This was done for two main reasons. First, for many people, families are the bedrock of identity and wellbeing [50]. Family members and partners play a crucial role in resilience by supporting, believing in, and encouraging family members [12,49]. Family systems are said to mediate and regulate individual vulnerability and

the impact of adverse events. Relationships with kin, intimate partners and mentors play a crucial role in resilience [12,48,49]. And second, stigma and discrimination have an impact on the whole family. In addition, a family-based intervention can be practiced at home and no travel is required, and is therefore relatively inexpensive (compared to e.g. peer support groups or counselling).

## Impact of intervention on resilience

The intervention in the present study significantly improved resilience scores in participants from Odisha state post-intervention, but not in participants from Telangana state. In Odisha, this improvement was maintained at six-month follow-up. The baseline score of the participants in Odisha and Telangana state in the present study, were similar to that of elderly patients with depression in a study in Maharashtra state in India [51]. Follow-up scores in Odisha state were similar to that of healthy, elderly controls in the same study [51], but somewhat higher than the scores found in two cross-sectional studies in Karnataka state among parents of children with intellectual disability and adult offspring of parents with schizophrenia [52,53].

Several interventions aiming to improve resilience and wellbeing have found increases in CD-RISC scores, with most reporting an increase of around 15% [54]. The present study found no increase in resilience in Telangana state and 66% increase in Odisha state. This is a much higher increase than similar studies have reported [54], Indeed there have also been several studies who found no increase [55,56] or a decrease on the CD-RISC after intervention [57,58].

We surmised that this substantial discrepancy may in part be attributable to differences in context and community demographics [59,60]. Community ties, opportunities for participation in community life, opportunities to connect with others, as well as religious or other groups, can affect resilience and family functioning [61]. It is likely that community bonds were stronger in the rural areas of Odisha state as opposed to the Telangana participants, who were mostly extremely poor urban slum residents (almost half of the participants from Telangana state were unemployed). In addition, Hinduism in Telangana state was associated with more improvements in resilience and quality of life post-intervention. The controversial 'Citizenship Amendment Act' that was passed by the Parliament of India in December 2019, that offers citizenship to non-Muslims fleeing religious persecution from nearby countries, and related protests and demonstrations likely also had a negative impact on the Muslim participants of Telangana state. This is an area for future exploration. Importantly, the intervention also coincided with the COVID-19 pandemic and restrictions in Telangana (which were not as severe in Odisha). It is possible that the psychosocial nature of the intervention was not substantial enough to address their more fundamental and overwhelming needs. On the other hand, the Odisha participants were mostly tribal people, for whom such interventions were entirely novel. Further, the effects of the pandemic were less substantial and had not yet penetrated their region at the time of follow-up. It is possible that this was the first psycho-social intervention they had ever experienced and therefor drew considerable benefit. Finally, given the crucial role of social support in fostering resilience [12,47–49], it is possible that the participants from Odisha state had stronger social relationships and support prior to the intervention.

## Impact of intervention on quality of life

The intervention in the present study significantly improved the quality of life of persons affected from both states, and of family members from Odisha state. In Odisha, where we

conducted a second follow-up assessment, the improvement in quality of life remained six months after the intervention. This is an important finding, given that leprosy and leprosy-related stigma can have a negative impact on quality of life [36,62,63]. Studies have shown that resilience and quality of life are positively correlated [64,65]. Our results confirm the mediating role of resilience and social support on quality of life found in other studies [16,66,67].

The improvement in quality of life noted for people affected, but not for their family members, in Telangana suggests that the attention provided to them through the project may have had substantial benefit for them, but as noted above, was insufficient to improve the quality of life of their family members. It is self-evident that improving the resilience of families in extreme poverty requires more of a long-term, multifaceted and systemic intervention. This is an important lesson for future research.

### Study limitations

As noted in the discussion, our study was substantially affected by the COVID-19 pandemic, which took effect in India while the intervention was underway. COVID-19 affected our data collection (limited follow-up in Telangana), and also appears to have adversely affected our outcome measure scores, but the exact nature and extent of that influence is a matter of conjecture. Suffice to say that we took confidence in the appeal of our intervention in that no families dropped out of the pilot study, and in the qualitative feedback, all described drawing benefit from the programme.

Conducting the project across multiple languages and cultures was also a substantial limitation that may have constrained comprehension at some points. In both cohorts there were a number of local dialects/languages, however all participants understood the language of intervention and outcome measurement (Urdu in Telangana, and Odia in Odisha). This limitation was to some extent mitigated by using local staff who were familiar with the families and the local dialects/languages. Further, while our own internal translation of the CD-RISC into Odia was certainly beneficial, it was not validated in that language, so results should be considered cautiously. Indeed the discrepancies of scores across cohorts may in part be attributable to language and translation concerns.

We recommend including pre- and post-intervention assessment of knowledge and community stigma in future studies, in order to ensure a better understanding of the changes in knowledge and community stigma. For future studies it would also be worthwhile to assess internalised stigma, mental wellbeing, physical health and stressor exposure also, to explore what role these factors play in strengthening resilience and interventions to strengthen resilience. The relationship between resilience and religion is another area for future exploration. As noted above, since this pilot study included relatively small numbers of participants across various religious groups, and coincided with religious persecution due to the 'Citizenship Amendment Act', interpretation of these data must be cautious. However, in response to these indications and current literature on this topic [11], there is great need for more detailed exploration of the interplay between resilience and religion, beliefs, spirituality and faith practices.

### Conclusions

This pilot study showed that the 10-week family-based intervention to strengthen resilience among persons affected by leprosy and their family members is feasible, and has the potential to improve resilience and quality of life. This is one of the first interventions designed to strengthen psychosocial resilience of persons affected by NTDs such as leprosy in a developing

country context. A large-scale efficacy trial is necessary to determine the effectiveness and long-term sustainability of the intervention.

## Supporting information

**S1 STROBE Checklist.**
(DOC)

**S1 Text. Overview of sessions in the resilience intervention and corresponding dimensions of resilience.**
(DOCX)

**S2 Text. Difference per question pre- and post-intervention on the resilience scale (CD-RISC), the quality of life scale (WHOQOL-BREF) and in raw scores per domain on the WHOQOL-BREF.**
(DOCX)

**S3 Text. Factors associated with more short-term improvement on the resilience scale (CD-RISC) and the quality of life scale (WHOQOL-BREF).**
(DOCX)

## Acknowledgments

We are grateful to the contributions and trust of the participants. We thank the research team in Telangana and Odisha state who facilitated the intervention and collected the data. We are grateful to the support and advice of the ILEP Advisory Panel of women and men affected by leprosy: Mathias Duck, Paula Soares Brandão, Rachna Kumari, Kofi Nyarko and Amar Timalsina. We want to thank and acknowledge Monty Mukhier for the financial management of this study. Finally, we want to thank Dr Wim van Brakel of NLR for his statistical advice.

## Author Contributions

**Conceptualization:** Anna T. van't Noordende, Zoica Bakirtzief da Silva Pereira, Pim Kuipers.

**Data curation:** Anna T. van't Noordende, Pritha Biswas, Mohammed Ilyas, Jayaram Parasa.

**Formal analysis:** Anna T. van't Noordende, Pim Kuipers.

**Funding acquisition:** Anna T. van't Noordende, Zoica Bakirtzief da Silva Pereira, Pim Kuipers.

**Investigation:** Anna T. van't Noordende, Pritha Biswas, Mohammed Ilyas, Vijay Krishnan, Jayaram Parasa, Pim Kuipers.

**Methodology:** Anna T. van't Noordende, Zoica Bakirtzief da Silva Pereira, Pim Kuipers.

**Project administration:** Anna T. van't Noordende, Pritha Biswas, Mohammed Ilyas, Vijay Krishnan, Jayaram Parasa, Pim Kuipers.

**Resources:** Anna T. van't Noordende, Zoica Bakirtzief da Silva Pereira, Pritha Biswas, Mohammed Ilyas, Vijay Krishnan, Jayaram Parasa, Pim Kuipers.

**Software:** Anna T. van't Noordende.

**Supervision:** Zoica Bakirtzief da Silva Pereira, Pritha Biswas, Mohammed Ilyas, Vijay Krishnan, Jayaram Parasa, Pim Kuipers.

**Validation:** Anna T. van't Noordende, Zoica Bakirtzief da Silva Pereira, Pim Kuipers.

**Visualization:** Anna T. van't Noordende.

**Writing – original draft:** Anna T. van't Noordende, Pim Kuipers.

**Writing – review & editing:** Anna T. van't Noordende, Zoica Bakirtzief da Silva Pereira, Pritha Biswas, Mohammed Ilyas, Vijay Krishnan, Jayaram Parasa, Pim Kuipers.

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
