## [Decision Letter · Decision Letter 0]

16 Feb 2021

Dear Ms. van 't Noordende,

Thank you very much for submitting your manuscript "Strengthening individual and family resilience against leprosy-related discrimination: a pilot intervention study" for consideration at PLOS Neglected Tropical Diseases. As with all papers reviewed by the journal, your manuscript was reviewed by members of the editorial board and by several independent reviewers. In light of the reviews (below this email), we would like to invite the resubmission of a significantly-revised version that takes into account the reviewers' comments. 

The manuscript deals with an important and little explored subject in leprosy. 

I recommend to the authors a detailed reading of the reviewers' comments and the accomplishment of the suggested modifications.

We cannot make any decision about publication until we have seen the revised manuscript and your response to the reviewers' comments. Your revised manuscript is also likely to be sent to reviewers for further evaluation.

Sincerely,

Susilene Maria Tonelli Nardi, Ph.D

Deputy Editor

Susilene Nardi

Deputy Editor

The manuscript deals with an important and little explored subject in leprosy. 

I recommend to the authors a detailed reading of the reviewers' comments and the accomplishment of the suggested modifications.

Reviewer's Responses to Questions

**Key Review Criteria Required for Acceptance?**

**Methods**

-Are the objectives of the study clearly articulated with a clear testable hypothesis stated?

-Is the study design appropriate to address the stated objectives?

-Is the population clearly described and appropriate for the hypothesis being tested?

-Is the sample size sufficient to ensure adequate power to address the hypothesis being tested?

-Were correct statistical analysis used to support conclusions?

-Are there concerns about ethical or regulatory requirements being met?

Reviewer #1: No.

Yes, however some modifications are needed.

No. It is necessary provide disability grade of treated participants, Multidrug therepy scheme whic will interfere in resilience analysis.

Yes. 

No.

Yes, however is needed the protocol number of approving.

Reviewer #2: Objective is quite clear and promising. Method (quasi-experimental) correctly selected for this purpose. The population has proper specificities that may restrict results to them or, of course, to social groups similar to them. However, the studied groups are clearly defined and described - therefore, no conflict arises from this particularity.

Reviewer #3: The objectives are clearly articulated. The authors have a clear direction as to what they intend to achieve as evident in their design and use of validated tools for assessing the outcomes. However, there are concerns as to why the authors decided to aggregate the scores of the individual domains of the WHO-QoLBREF tool as a measure of QoL. The scores from the QoL domains should be explored as intended by the tool.

**Results**

-Does the analysis presented match the analysis plan?

-Are the results clearly and completely presented?

-Are the figures (Tables, Images) of sufficient quality for clarity?

Reviewer #1: No.

No.

No.

Reviewer #2: (No Response)

Reviewer #3: The analysis plan is clear. However, it needs to be modified. The QoL tool used explores QoL in 4 dimensions as stated by the authors. It should not be modified without statistical reason; having been validated to explore QoL as intended.

**Conclusions**

-Are the conclusions supported by the data presented?

-Are the limitations of analysis clearly described?

-Do the authors discuss how these data can be helpful to advance our understanding of the topic under study?

-Is public health relevance addressed?

Reviewer #1: No.

Yes.

Yes.

Yes.

Reviewer #2: (No Response)

Reviewer #3: The authors state that the covid-19 pandemic was a limitation to the study. It is not exactly clear how this happened and the dimension of the effect of the work. The authors seem to say there was an effect on the accuracy of the findings because of this. This should either be revised or clearly shown how.

**Editorial and Data Presentation Modifications?**

Reviewer #1: (No Response)

Reviewer #2: (No Response)

Reviewer #3: Minor revision

**Summary and General Comments**

Reviewer #1: Introduction

No suggestions.

Methods

Participants and sampling procedure

From line 153 to 159 authors should provide further important information such as disability degree of treated leprosy cases and time after treatment and type of treatment (multibacillary or paucibacillary accordint to WHO), because those who have disabilities and some skin alterations will suffer stigma as compared with those 0 degree and that did not treat with Multibacillary schemes.

Authors should make clear that they used the participant as his/her own control along the different times of intervention.

From line 167 to 172 I agree with authors, however, for the subsequente study, the sample size should be calculated using a the effect size found in this pilot study through a statistics software. 

Ref: Moore CG, Carter RE, Nietert PJ, Stewart PW. Recommendations for planning pilot studies in clinical and translational research. Clinical and translational science. 2011;4(5):332-7.

Data collection

From line 175 to 180 authors should rewrite and changing some words, for instance, gender and education, since that sex and schooling are more appropriated. 

“Sex refers to the biological distinctions between males and females, most often in connection with reproductive functions. Gender, by contrast, emphasises the socially constructed differences between men and women that give rise to masculinity and femininity.”

Ref: Short SE, Yang YC, Jenkins TM. Sex, gender, genetics, and health. American journal of public health. 2013;103 Suppl 1:S93-101.

Regarding schooling, it refers to a formal process related to institution of education. I think authors are using this meaning for this word . 

From line 182 to 206 I recommend to authors to create a new subtitle called “Research questionnaires” or similar. From line 208 to 2014 I suggest to authors create a subtitle named “Phases of follow-up time” or similar for describing these different times of follow-up.

Data analysis

From line 219 to 231, authors wrote about the data analyses. I recommend them starting this section statint in relation to normality tests applied, since that questionaires which provide ordinary scores the figures have as a result non-parametric distribution. Thus, authors may use Wilcoxon for two medians or Friedman test for more than 2 medians when it deal with paired method.

Afterward, authors should quote the statistical tests utilised to found out these findings. I suggest the use of Binomial test for comaparing the difference between proportions showed in table 1. 

Authors should quote the manufacturer and version of statistical tests used in this analysis in a complete sentence (not only SPSS). 

From line 225 to 229 it was not clear what type of regression model was used in this analysis. Linear reagression accept only normal distribuition (it requeres normality test application). 

Did the authors use the Logistic Regression? The text must be written clearly.

Ethical considerations

Is there a protocol number or approving nunber for this research provided by Institutional Ethics Committee of the Lepra-Blue Peter Public Health and Research Centre?

Results

In accordance with data showed in table 1. Authors should use the binomial test for comparing differences between Odisha and Telangana states proportions. Whether authors are interested in prove the diferences between these two groups they have starting showing diferences among demografic informations. 

Are the the sex, religion, occupation associated with any group?

The binomial test or another association test will respond this question.

For each comparison authors should provide p-value.

What is the data regarding disability grade? It is very important to discuss about resilience.

Authors should provide data about type of treatment, whether multibacillary or paucibacillary. Clofazimine can cause ichthyosis and change in skin colour what will interfere in stigma and resilience of these individuals.

The standard deviation suggest that age data are non-parametric. Hence, the authors shall never use mean, however they should use median to represent the age.

According to table 2 and 3, I have noticed they are utilising the same data. Nevertheless, the table 3 compared three averages. Therefore, authors should make one table summarising the table 2 and 3. It is more accurate compare the 3 averages instead 2 averages (or difference of mean rank). 

Whether authors test the normality and consequently the data will be non-parametric, what I believe, they should use the Friedman test for comparing 3 or more follow-up times considering the participant as his/her own control. If the distribuition passed in normality test, mean and SD should be provided. Otherwise, authors must use median, maximum and minimum values found for each median.

On the other hand, I have noted that authors made a mistake using t-test not for the reason associated with normality test, however for the fact they compare 3 different follow-up times by means of t-test, since that it may represent a type I error. The comparison encompassing, 3 or more groups o follow-up times, paired, must be lead by use o Friedman or ANOVA paired if the distribuition is normal.

I recommend that authors should remake the data analyses, what will obligate them rewrite some parts of results and discussion.

From line 296 to 315, authors showed results about multivariate analysis by means of Regression model such as seen in supporting file 3.

As these findings deal with scores given throught a Questionnaire, I enphasise that normality data should be provided and generally one measure, represented by a score, is not a parameter for another, so it should be considered as a non-parametric variable.

The analysis in supporting file 3 did not show where authors find the R square for these data. If the model called by authors another occupation than paid work and participants who were Hindu can explain 24% of variability of increase in resilience for CD-RISC, the R square or correlation coefficient should be given by authors in supplementary data.

Reviewer #2: A most welcome study on stigma, discrimination and resilience in leprosy.

The selection of urban and rural area with a reasonable low detection rate was also quite opportune to reproduce a social environment that may resemble the average behavioral condition of averge villages in leprosy endemic countries anywhere. Studying areas with very high or close to zero detection rate would give non reproducible results to be widely applied. In addition, on should consider that, for a study on stigma/discrimination issues, it is interesting to note that 70% had DG2, that is, visible deformities, that is, they were prone to easily be recognized as leprosy patients. This grants value to the results of the study.

Also opportune was the inclusion of human rights issues in the action learning process, since to me it seems crucial as part of addressing resilience aspects of stigmatized people.

For practical purposes, the association of the bamboo and its vegetal characterizes to resilience concept was a marvelous decision. Indeed, participants acknowledged this.

Though one understand the basic and/or advanced differences between Muslim and Hindu creed, it is estimated that profession of these creeds in a same ethnological culture has some interweaving’s. This leads to the fact that religion is a relevant variable to be discussed in this study, what was done scarcely – unfortunately. As a matter of fact, from the article we learn that in Telangana state both CD-RISK and WHOQOL had significantly more improvement among Hindus between baseline and the first follow-up assessments. This finding required a more detailed discussion! In a future study, religion as a variable should be throughout explored.

The presentation of Results and Discussion may require some revision - please, a text length reduction applies. There are, to my personal feeling, excessive description of the findings of referred articles. This can be helpful to those not willingly to go for the entire cited article, but may render the reading of the original article tiresome. One example, but not the only: line 419. In Odisha, this (line 420) improvement was maintained at six-month follow-up. Singh and colleagues [51], who used the CD ....What follows is an almost full description of the referred article including figures!

There are other citations such as this along the Discussion that, to my view, should be restricted to the direct liaison with the author's sentence and limited to the reference. Details of the referred article should not necessarily by reproduced within the article. Or, at least, make a selection of those references you find truly necessary.

Reviewer #3: Study addresses and articulates what it intends to solve.

PLOS authors have the option to publish the peer review history of their article (what does this mean?). If published, this will include your full peer review and any attached files.

Reviewer #1: No

Reviewer #2: No

Reviewer #3: Yes: EMAEDIONG IBONG AKPANEKPO
---

## [Editor Report · Decision Letter 1]

25 Mar 2021

Dear Ms. van 't Noordende,

We are pleased to inform you that your manuscript 'Strengthening individual and family resilience against leprosy-related discrimination: a pilot intervention study' has been provisionally accepted for publication in PLOS Neglected Tropical Diseases.

Best regards,

Susilene Maria Tonelli Nardi, Ph.D

Deputy Editor

Susilene Nardi

Deputy Editor

---

## [Editor Report · Acceptance letter]

30 Mar 2021

Dear Ms. van 't Noordende,

We are delighted to inform you that your manuscript, "Strengthening individual and family resilience against leprosy-related discrimination: a pilot intervention study," has been formally accepted for publication in PLOS Neglected Tropical Diseases.

Best regards,

Shaden Kamhawi

co-Editor-in-Chief

Paul Brindley

co-Editor-in-Chief
